# Application of Graphene Oxide in Oral Surgery: A Systematic Review

**DOI:** 10.3390/ma16186293

**Published:** 2023-09-20

**Authors:** Francesco Inchingolo, Angelo Michele Inchingolo, Giulia Latini, Giulia Palmieri, Chiara Di Pede, Irma Trilli, Laura Ferrante, Alessio Danilo Inchingolo, Andrea Palermo, Felice Lorusso, Antonio Scarano, Gianna Dipalma

**Affiliations:** 1Interdisciplinary Department of Medicine, University of Bari “Aldo Moro”, 70124 Bari, Italy; francesco.inchingolo@uniba.it (F.I.); angeloinchingolo@gmail.com (A.M.I.); dr.giulialatini@gmail.com (G.L.); giuliapalmieri13@gmail.com (G.P.); c.dipede1@studenti.uniba.it (C.D.P.); trilliirma@gmail.com (I.T.); lauraferrante79@virgilio.it (L.F.); giannadipalma@tiscali.it (G.D.); 2College of Medicine and Dentistry, Birmingham B4 6BN, UK; andrea.palermo2004@libero.it; 3Department of Innovative Technologies in Medicine and Dentistry, University of Chieti–Pescara, 66100 Chieti, Italy; drlorussofelice@gmail.com (F.L.); ascarano@unich.it (A.S.)

**Keywords:** graphene oxide (GO), dental surgery, graphene coating, oxide materials

## Abstract

The current review aims to provide an overview of the most recent research in the last 10 years on the potentials of graphene in the dental surgery field, focusing on the potential of graphene oxide (GO) applied to implant surfaces and prosthetic abutment surfaces, as well as to the membranes and scaffolds used in Guided Bone Regeneration (GBR) procedures. “Graphene oxide” and “dental surgery” and “dentistry” were the search terms utilized on the databases Scopus, Web of Science, and Pubmed, with the Boolean operator “AND” and “OR”. Reviewers worked in pairs to select studies based on specific inclusion and exclusion criteria. They included animal studies, clinical studies, or case reports, and in vitro and in vivo studies. However, they excluded systematic reviews, narrative reviews, and meta-analyses. Results: Of these 293 studies, 19 publications were included in this review. The field of graphene-based engineered nanomaterials in dentistry is expanding. Aside from its superior mechanical properties, electrical conductivity, and thermal stability, graphene and its derivatives may be functionalized with a variety of bioactive compounds, allowing them to be introduced into and improved upon various scaffolds used in regenerative dentistry. This review presents state-of-the-art graphene-based dental surgery applications. Even if further studies and investigations are still needed, the GO coating could improve clinical results in the examined dental surgery fields. Better osseointegration, as well as increased antibacterial and cytocompatible qualities, can benefit GO-coated implant surgery. On bacterially contaminated implant abutment surfaces, the CO coating may provide the optimum prospects for soft tissue sealing to occur. GBR proves to be a safe and stable material, improving both bone regeneration when using GO-enhanced graft materials as well as biocompatibility and mechanical properties of GO-incorporated membranes.

## 1. Introduction

The landscape of modern healthcare has been continuously shaped by groundbreaking advancements in materials science and technology [1]. One such revolutionary material that has garnered significant attention is graphene, a two-dimensional carbon allotrope characterized by its exceptional properties [2,3,4]. It is represented by a hexagonal honeycomb structure in which each atom is able to bond with three adjacent atoms (Figure 1) [5,6,7]. Graphene’s unrivaled mechanical strength, electrical conductivity, and biocompatibility have sparked interest across various scientific domains. In the realm of oral surgery, where precision, efficacy, and patient well-being are paramount, the integration of graphene holds tremendous promise for pushing the boundaries of traditional approaches and ushering in a new era of surgical innovation [8,9].

Promising uses for graphene include lots of biomedical fields [10]. Graphene’s electrical conductivity, high specific surface area, and mechanical robustness can help transdermal biosensors provide signals with greater precision and repeatability to monitoring molecules and biomarkers [11]. Graphene has several potential uses as a consequence, including drug delivery systems, and has attracted a lot of interest in the field of biomedical 3D printing [10,12]. It has recently been demonstrated his impact in neurotherapeutics for neuroimaging, neuro-oncology, and neuro-surgery [13].

Oral surgery encompasses a spectrum of procedures, ranging from routine tooth extractions to complex maxillofacial reconstructions [8,14,15,16,17]. The quest for enhanced patient outcomes and the refinement of surgical techniques has been an ongoing pursuit in this field [18]. Graphene, with its unique attributes, emerges as a material with the potential to redefine the landscape of oral surgery, offering novel solutions for challenges that have long persisted and introducing avenues for previously unexplored possibilities [19].

At the heart of graphene’s allure lies its remarkable physical properties. Structurally, graphene consists of a single layer of carbon atoms arranged in a hexagonal lattice [20]. This arrangement imparts extraordinary mechanical strength, rendering graphene the strongest material ever tested. Such mechanical robustness holds promise for oral surgery, where materials capable of withstanding physiological forces while promoting integration with surrounding tissues are highly sought after [21]. Graphene’s strength can be harnessed in the fabrication of dental implants, orthodontic devices, and reconstructive scaffolds that maintain structural integrity and support tissue regeneration [22].

Graphene’s electrical conductivity is equally intriguing. Its high electron mobility opens the door to applications involving electrical stimulation and biosensing. In the context of oral surgery, this property could lead to the development of implantable devices capable of monitoring healing processes in real time, thereby enabling timely interventions in case of complications [23]. Additionally, the integration of graphene-based sensors could enhance the accuracy of surgical procedures, offering surgeons immediate feedback and aiding in precise tissue manipulation [24].

A cornerstone of successful oral surgery is the interaction between surgical materials and the complex biological environment [25]. In-depth research has been conducted on the biocompatibility of graphene, and studies have shown that it may be able to facilitate cellular adhesion, proliferation, and differentiation [26]. The promise of graphene as a scaffold material for tissue engineering applications in oral surgery is highlighted by this feature. Graphene scaffolds have the potential to speed up the creation of bioengineered oral tissues, bone regeneration, and wound healing by creating an environment that is favorable for cellular growth [27].

Moreover, graphene’s interactions with immune cells have raised intriguing possibilities for modulating immune responses during surgical interventions [28,29]. This presents the potential to reduce inflammation, enhance tissue integration, and ultimately improve patient recovery and comfort post-surgery [28].

The prevention of post-operative infections remains a critical challenge in oral surgery [30]. Graphene’s inherent antibacterial properties have attracted considerable attention. Its unique interaction with bacterial cell membranes disrupts their structural integrity, rendering them susceptible to elimination [31,32]. This property could be leveraged to develop antimicrobial coatings for surgical instruments, implants, and wound dressings [33,34,35]. By mitigating bacterial colonization and biofilm formation, graphene-based materials could substantially reduce the risk of infections, leading to improved patient outcomes and decreased reliance on antibiotics [36,37].

While the potential applications of graphene in oral surgery are undeniably exciting, several challenges must be addressed to ensure safe and effective clinical implementation. The scalable synthesis of graphene materials suitable for surgical use, long-term biocompatibility assessments, and regulatory approvals are among the foremost challenges. Additionally, the development of standardized surgical protocols and techniques for incorporating graphene into existing procedures is essential to ensure seamless integration and optimal outcomes [23,38].

As the frontiers of materials science and oral surgery intersect, graphene emerges as a transformative force poised to reshape the landscape of surgical practices [34]. Its extraordinary mechanical, electrical, and biocompatible properties offer novel solutions to age-old challenges while presenting unprecedented opportunities for innovation [39,40]. This comprehensive exploration sheds light on the manifold applications of graphene in oral surgery, emphasizing the potential to revolutionize patient care, surgical techniques, and the overall trajectory of the field [41,42]. As researchers and clinicians continue to unravel graphene’s potential, the future of oral surgery appears brighter and more promising than ever before [43,44].

## 2. Materials and Methods

### 2.1. Protocol and Registration

This systematic review was conducted by the standards of the Preferred Reporting Items for Systematic Reviews and Meta-analysis (PRISMA), and it was submitted to PROSPERO with number ID 453609.

### 2.2. Search Processing

Graphene oxide, dental surgery, and dentistry were the search terms utilized on the databases (Scopus, Web of Science, and Pubmed) to select the papers under evaluation, with the Boolean operators “AND” and “OR”.

The search was restricted to just items released in English during the previous ten years (July 2013–July 2023).

### 2.3. Eligibility Criteria

The reviewers, who worked in pairs, chose works that satisfied the following criteria for inclusion: (1) animal studies; (2) clinical studies or case reports; and (3) in vitro and in vivo studies.

Exclusion criteria were systematic reviews, narrative reviews, and meta-analyses.

### 2.4. Data Processing

The screening procedure, which was carried out by reading the article titles and abstracts chosen in the earlier identification step, allowed for the exclusion of any publications that varied from the themes looked at.

The complete text of publications that had been determined to match the predetermined inclusion criteria was then read.

Reviewer disagreements on the choice of the article were discussed and settled.

### 2.5. Quality Assessment

The quality of the included papers was assessed by two reviewers, RF and EI, using the reputable Cochrane risk-of-bias assessment for randomized trials (RoB 2). The following six areas of possible bias are evaluated by this tool: random sequence generation, allocation concealment, participant and staff blinding, outcome assessment blinding, inadequate outcome data, and selective reporting. A third reviewer (FI) was consulted in the event of a disagreement until an agreement was reached.

## 3. Results

Keyword searches of the Web of Science (55), Scopus (38), and Pubmed (200) databases yielded a total of 293 articles.

The subsequent elimination of duplicates (61) resulted in the inclusion of 232 articles.

Of these 232 studies, 213 were excluded because they deviated from the previously defined inclusion criteria.

The screening phase ended with selecting 19 publications for this work.

The results of each study are reported in Figure 2.

The study data was selected by analyzing the study design, number of patients, intervention, and outcomes (Table 1).

## 4. Discussion

### 4.1. Implant and Abutment surfaces

This review discusses the potential of graphene oxide (GO) as a promising nanomaterial with exceptional physical and chemical properties. Recent research has focused on its applications in biomedical fields such as tissue engineering, antimicrobial materials, and implants [64]. The review examines the use of graphene to functionalize dental implant surfaces and its interactions with host tissue. Graphene is a single layer of sp2 hybridized carbon atoms arranged in a honeycomb lattice, known for its remarkable strength, elasticity, and electrical characteristics [34,65]. GO and reduced graphene oxide (rGO) are its primary derivatives. Due to their biocompatibility, low toxicity, hydro-solubility, and reactive oxygen groups, studies suggest that graphene and GO can support tissue regeneration, cell differentiation, and proliferation [1,2,3]. They also enhance the bioactivity and mechanical performance of biomaterials and can serve as carriers for drugs and biomolecules [6,54,65,66,67,68,69].

In order to manage infection and stop bone loss, peri-implantitis therapy must remove polymicrobial biofilms from the implant site and lessen tissue invasion. Brushing and GO at high concentrations effectively decontaminate biofilms from exposed titanium surfaces, as shown by Qin et al. [53].

The study by Ren et al. utilized GO as a coating on titanium foils to deliver drugs and enhance the growth and differentiation of rat bone mesenchymal stem cells (rBMSCs) into osteoblasts (Figure 3) [46]. The researchers incorporated dexamethasone (DEX) onto GO-coated titanium implants, resulting in improved absorption and sustained release of the drug. The DEX-GO-Ti substrates showed higher rBMSC proliferation compared to control and DEX-rGO-coated substrates [46]. Moreover, rBMSCs exhibited enhanced osteogenic differentiation on DEX-GO-Ti and DEX-rGO-Ti surfaces. This approach effectively controlled the bioactivity of titanium implants, showing promise for advancements in dentistry applications [46].

This study focuses on enhancing the antibacterial and cytocompatible attributes of titanium alloy implants by employing electrophoretic deposition to create bioglass (BG)-GO composites [45]. The resultant BG-GO composites formed a consistent and dense coating layer, measuring 50–55 μm in thickness [45]. This coating displayed enhanced resistance against corrosion and heightened antibacterial efficacy in comparison to samples coated solely with BG [45]. This antibacterial effectiveness escalated with an increase in GO content. Cell adhesion findings indicated favorable biocompatibility of the BG-GO composite coating. Furthermore, the inclusion of GO in the BG-GO coating did not impede cell attachment to the alloy sample [45]. Consequently, the electrophoretic deposition method for creating BG-GO composite coatings with these beneficial traits presents a promising alternative for bone implant applications [45].

The physiochemical properties of GO-carbon fibers (CF)-PEEK on titanium implants were analyzed by Qin et al., who revealed that these coatings might greatly reduce the coefficient of friction of alloy and improve wear resistance [51]. In addition, GO-CF-PEEK showed biological safety and improved osteointegration [52].

Photothermal therapy (PTT), an alternative antibacterial treatment, has a substantial impact on deactivating oral microbiota. The study by Park et al. involved applying graphene possessing photothermal characteristics onto a zirconia surface using atmospheric pressure plasma, followed by an assessment of its antibacterial effects against oral bacteria [47]. To coat graphene oxide onto zirconia specimens, an atmospheric pressure plasma generator (PGS-300, Expantech, Suwon, Republic of Korea) was employed, utilizing an Ar/CH4 gas mixture at a power of 240 W and a flow rate of 10 L/min [47]. Notably, the group subjected to near-infrared irradiation after coating the zirconia specimen with graphene oxide exhibited a significant decrease in *S. mutans* and *P. gingivalis* adhesion compared to the non-irradiated group [47]. This reduction in oral microbiota activity was attributed to the photothermal effect on the zirconia surface coated with graphene oxide, which demonstrated photothermal properties [47].

Cheng et al. analyzed a new type of coating using a combination of GO and the antimicrobial peptide (AMP) Nal-P-113 on a smooth titanium surface [48]. The study evaluates the effectiveness of this coating at fighting bacteria and whether it is compatible with cells. The findings revealed that Nal-P-113 was gradually released from the composite coating over time when tested in a lab setting [48]. The GO coating loaded with Nal-P-113 demonstrated strong antibacterial properties against both Streptococcus mutans and Porphyromonas gingivalis, with no noticeable harm to human gingival fibroblast (HGF) cells [48]. However, further refinement is necessary to optimize the Nal-P-113-loaded GO coating for its potential to prevent infection and promote healing in the tissues surrounding implants [48]. The same results were obtained by Guo et al., who analyzed GO-modified Polyetheretherketone (PEEK) implant abutment surfaces grafted with dopamine [50].

The study by Jang et al. aimed to investigate the impact of applying GO onto a zirconia (Zr) surface on bacterial bonding and osteoblast activation [49]. Two groups of zirconia samples were compared: one without coating (Zr control) and another with GO coating (Zr-GO) [49]. Analysis through a scanning electron microscope confirmed successful GO deposition on the Zr-GO group. *S. mutans* bacterial attachment and growth were significantly reduced on the Zr-GO surface, while the attachment of MC3T3-1 cells remained similar, but their growth and specialization improved on Zr-GO compared to Zr [49].

Conclusively, GO-coated zirconia hindered *S. mutans* bacterial attachment and promoted osteoblast growth and specialization, suggesting a potential prevention of peri-implantitis by deterring bacterial adhesion and enhancing implant success by improving bone attachment [49].

### 4.2. Scaffolds and Membranes

Recently, scientific research on graphene has focused on regenerative techniques in oral surgery, going on to investigate the efficacy of graphene oxide added to membranes or scaffolds compared with conventional methods with the hope that the results, combined with the potential of stem cells, will lead to a new class of nanomaterials with unique properties and a significant impact in the field of nanotechnology and oral health.

For example, the evaluation of a bone graft material consisting of biphasic calcium phosphate (BCP) coated with rGO was the focus of the investigation by Jeong-Woo Kim et al. Osteoblast viability decreased as the concentration of rGO nanoplates increased in terms of cytotoxicity, with significant decreases at higher concentrations, while new bone production dramatically increased compared with the control group in in vivo tests using rat calvarial lesions. In fact, according to micro-CT and histomorphometric evaluations, rGO-coated BCP groups had higher volumes and percentages of new bone. The rGO4 group (with a 4:1000 ratio of rGO to BCP) showed the highest bone volume, demonstrating that the concentration of rGO in the composite material is important for bone regeneration [55].

In contrast, Erika Nishida et al. investigated the effects of adding a GO monolayer solution to a three-dimensional collagen scaffold for possible use in bone tissue engineering [56]. A special technique was used to produce the GO solution, and the resulting monolayer had an average width of about 20 m and a thickness of less than 1 nm. Next, the GO solution was mixed with a special solvent to obtain GO dispersions at various concentrations. The GO-modified scaffolds were injected into collagen scaffolds, and their different properties were evaluated. When the GO-modified scaffolds were characterized, they were found to have better physical characteristics, such as greater resistance to compression and enzymatic degradation, as well as a greater ability to adsorb calcium ions and proteins. To evaluate the effects of the modified GO scaffold, in vivo tests were performed on dog extractive cavities and rat subcutaneous tissues. The results showed that in rat tissues, the GO-modified scaffold stimulated angiogenesis and growth of cells and tissues and that compared with collagen-only scaffolds, the GO-modified scaffold significantly improved bone growth in dog extractive cavities. Overall, the research points to collagen scaffolds as attractive options for bone tissue engineering applications, as the addition of GO can improve their physical characteristics, cytocompatibility, and ability to form bone [56].

The research results of Jong Ho Lee et al. suggest that composite nanoparticles of hydroxyapatite (HAp) and rGO have extraordinary potential in enhancing the proliferation and osteogenic differentiation of pre-osteoblastic MC3T3-E1 cells.

In particular, extracellular calcium deposition in MC3T3-E1 cells was significantly enhanced by rGO/HAp composite nanoparticles, and clearly, calcium accumulation is a key sign of bone tissue creation and extracellular mineralization, two crucial steps of bone regeneration. This implies that composite nanoparticles could create a favorable environment for bone tissue development, thereby promoting osteogenesis. In addition, the enzymatic activity of alkaline phosphatase (ALP), an early indicator of osteogenic differentiation, was significantly elevated in the presence of composite nanoparticles. This shows that composite nanoparticles can accelerate the differentiation process of pre-osteoblastic cells.

The presence of composite nanoparticles also had a favorable impact on the deposition of osteogenic proteins, including osteopontin (OPN) and osteocalcin (OCN), which are essential markers of osteogenic cell development, suggesting a favorable impact on the expression of proteins important for bone growth. Therefore, these results also appear to be on the same wavelength as the other studies mentioned [57].

The work of Izumi Kanayama et al. [58] investigated the synthesis and characterization of GO and rGO films. Atomic force microscopy (AFM), scanning electron microscopy (SEM), and X-ray diffraction (XRD) were used to analyze the film morphology and tissue alterations of GO and rGO. Investigations on the biological characteristics of GO and rGO films were also conducted: the films were applied to culture plates and used by MC3T3-E1 mouse osteoblastic cells as substrates. From the results, it was found that the behavior of cells is affected differently by GO and rGO films. Compared with GO films, rGO films showed better cell activity. The films were also used to modify collagen scaffolds, resulting in improved tissue growth and compressive strength. Giant cells were present, and the materials and immune cells interacted favorably, indicating strong biocompatibility and a greater ability to stimulate cell activity and tissue integration. These results highlight the promising tissue engineering applications of GO and rGO and their ability to modify scaffolds to improve mechanical strength and tissue regeneration [58].

In the work of Chingis Daulbayev et al. [59], a composite was made by combining GO and HAp with a matrix of polycaprolactone (PCL), using an electrospinning technique, and the antibacterial capabilities of the composite on Gram-positive (*S. aureus*) and Gram-negative (*E. coli*) bacteria were analyzed; the antibacterial action of the composite was significant, and a larger clean zone was observed for higher concentrations of GO in the composite. The biocompatibility of the GO/HAp/PCL composite was also evaluated using MC3T3-E1 preosteoblast cells. Cell viability studies showed that the cytotoxic effects of the composite were minimal at lower concentrations: the cytotoxicity caused by HAp seemed to be attenuated by the addition of GO to the composite structure. It is also possible that GO has the ability to promote cell development, as it increases cell attachment and proliferation when added to the composite. Ultimately, the work showed that it was possible to successfully synthesize a GO/HAp/PCL composite with promising antibacterial qualities and biocompatibility [70,71,72].

In recently published work by Milena Radunovic et al. [60], the effects of GO-coated collagen membranes for guided bone regeneration (GBR) applications on dental pulp stem cells (DPSCs) were examined [73]. The research showed that attachment, proliferation, and osteogenic differentiation of DPSCs were promoted by the 2 and 10 g/mL GO-coated membranes, with a particular increase in metabolic activity, especially at higher concentrations. The fact that the cells adhered to the membrane surface without penetrating it was confirmed by hematoxylin-eosin staining. In addition, bone morphogenetic protein 2 (BMP2) and runt-related transcription factor 2 (RUNX2), indicators of osteogenic development, were significantly increased on GO-coated membranes, according to gene expression analysis. GO coating also significantly increased the secretion of prostaglandin E2 (PGE2), a crucial modulator of osteoblastic differentiation. On the other hand, with regard to inflammatory markers, tumor necrosis factor (TNF) and cyclo-oxygenase 2 (COX2) were downregulated, indicating a reduction in inflammation.

The study concluded that GO-coated collagen membranes limit inflammatory processes and promote attachment, proliferation, and osteogenic differentiation of DPSCs, emphasizing that efficacy is dose-dependent (the 10 g/mL concentration of GO produces the best results) [60].

The study presented by Letizia Ferroni et al. [61] investigated the creation and evaluation of rGO-PCL (reduced graphene oxide-polycaprolactone) composites for possible use in bone tissue engineering. Evaluation of the antibacterial efficacy of the composites against various bacterial strains revealed that they had a bacteriostatic effect on Gram-positive bacteria, particularly the 5% rGO-PCL composite. The study of the adhesion, morphology, and proliferation of human adipose-derived mesenchymal stem cells (HMSCs) was carried out on the surfaces of rGO-PCL, bringing out an upregulation of adhesion molecules, extracellular matrix elements (ECM) and metalloproteinases, indicating favorable cell-matrix interactions. In addition, the ability of HMSCs to differentiate into osteoblasts on the surfaces of rGO-PCL was examined. ALP activity and mineral matrix deposition were found to be maximal on the surface of 5% rGO-PCL, indicating better osteogenic differentiation capacity, and the expression of osteogenic markers such as OPN, OCN, RUNX2, and osterix (OSX) was found to be high on the surface of 5% rGO-PCL, indicating excellent osteoblastic proliferation. Certainly, therefore, the 5% rGO-PCL composite has proven to be a viable option for creating improved biomaterials for bone regeneration due to its demonstrated biocompatibility, bacteriostatic action against Gram-positive bacteria, and ability to enhance osteogenic differentiation [61].

In the very recent study by Elham-Sadat Motiee et al. [62] in 2023, a poly-3-hydroxybutyrate-chitosan (PC) scaffold reinforced with GO through the electrospinning method was developed with the aim of evaluating the fiber diameter, heat capacity, surface hydrophilicity, mechanical properties, and degradation of the nanocomposite scaffolds. It was again found that the above values are improved, suggesting that the improved characteristics and interactions of Poly-3 hydroxybutyrate-chitosan (PC)-GO nanocomposite scaffolds with cells and minerals may be promising for use in bone tissue engineering. In particular, the inclusion of GO improved the deposition of calcium and phosphate ions, indicating accelerated biomineralization, as well as increased cell adhesion, proliferation, and ALP activity, resulting in improved cell attachment, viability, and osteogenic activity [62].

The objective of the study by Alana P C Souza et al. [63] was to develop chitosan-xanthan-based membranes that also contained HAp and GO for potential use in guided bone regeneration (GBR) and again, as in the aforementioned studies, the results are extremely positive: in vitro bioactivity tests showed that HAp and GO increased bioactivity and promoted apatite deposition, and in particular higher concentrations of GO in membranes produced superior results in terms of cell viability, indicating increased cell adhesion and proliferation, which are essential for regenerative processes. Although the addition of particles did not improve mechanical properties, research on tensile strength showed that the membranes still exhibited qualities suitable for use as barriers and structural support in bone tissue regeneration [63].

### 4.3. Quality Assessment and Risk of Bias

The risk of bias in the included studies is reported in Figure 4. Regarding the randomization process, one study presents a high risk of bias and allocation concealment. All other studies ensure a low risk of bias. Only one study excludes performance; two studies confirm an increased risk of detection bias (self-reported outcome), and two of the included studies present a low detection bias (objective measures) (Figure 4). Two studies ensure a low risk regarding attrition and reporting bias.

## 5. Conclusions

The field of graphene-based engineered nanomaterials in dentistry is expanding due to their superior mechanical properties, electrical conductivity, and thermal stability. Graphene and its derivatives can be functionalized with bioactive compounds and added to dental materials, enhancing their properties. These materials stimulate tissue regeneration, cell differentiation, and proliferation while being biocompatible and low in toxicity. Dental implants and abutments with graphene coatings exhibit improved cytocompatibility, antibacterial properties, and osteoblast growth.

Graphene-enhanced scaffolds and membranes for guided bone regeneration (GBR) also have improved physical properties, leading to enhanced bone formation. Laboratory tests indicate increased secretion of osteogenic markers and reduced inflammatory markers on graphene-coated materials. However, there are concerns about the safety of graphene and its derivatives, necessitating further research to understand their long-term effects. Overall, graphene materials hold great potential for improving oral surgery procedures in the future.

## Figures and Tables

**Figure 1 materials-16-06293-f001:**
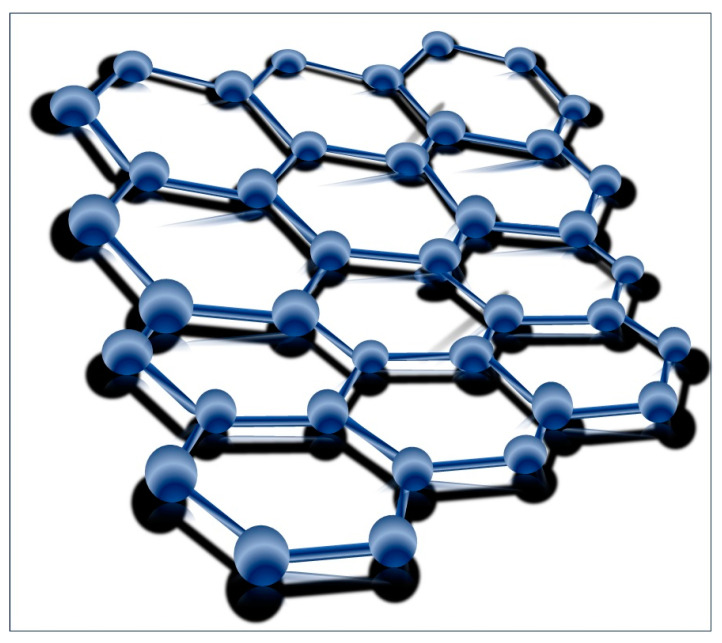
Crystal lattice of graphene with a hexagonal honeycomb structure in which each atom is able to bond with three adjacent atoms.

**Figure 2 materials-16-06293-f002:**
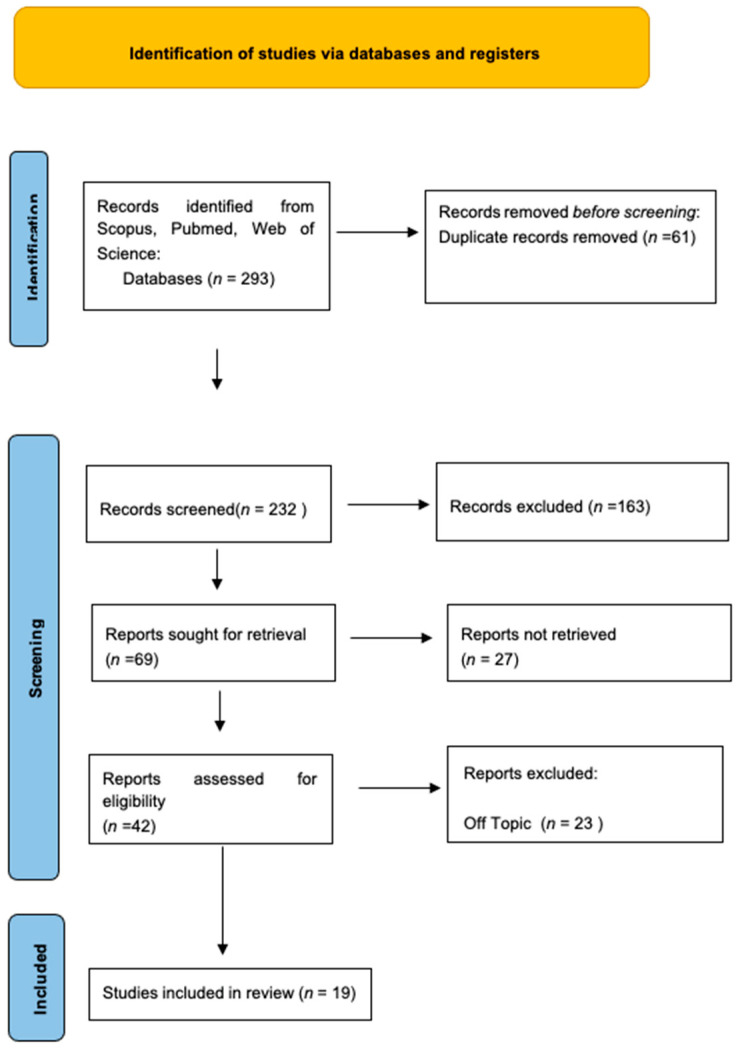
PRISMA flowchart.

**Figure 3 materials-16-06293-f003:**
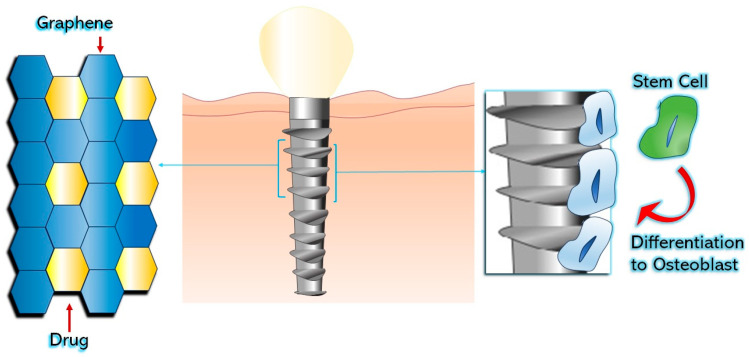
GO-coated implant surface enhances the differentiation of bone mesenchymal stem cells.

**Figure 4 materials-16-06293-f004:**
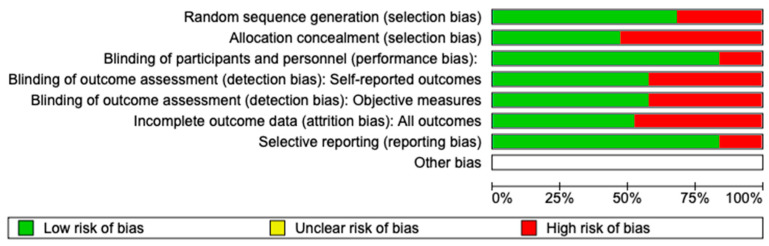
Risk of bias; red indicates high risk, and green indicates low risk of bias [45,46,47,48,49,50,51,52,53,54,55,56,57,58,59,60,61,62,63].

**Table 1 materials-16-06293-t001:** Characteristics of the studies included in the analysis.

Authors (Year)	Type of the Study	Aim of the Study	Materials	Results
Eshghinejad et al. [45] 2019	In vitro study	This article details our research into the electrophoretic deposition of composite materials consisting of BG-GO onto titanium alloy implants, aiming to enhance their antibacterial capabilities and biocompatibility.	Comparison of samples coated with BG-GO versus BG alone.	Enhanced antibacterial performance was observed in BG-GO-coated samples compared to BG-only coatings, with improved effectiveness as GO content increased. The BG-GO composite coating demonstrated favorable biocompatibility based on cell adhesion tests, indicating that the presence of GO did not hinder cell attachment to the alloy surface. Consequently, the BG-GO composite coatings, fabricated using the EPD technique and exhibiting these attributes, hold significant promise as a viable option for bone implant applications.
Ren et al. [46] 2019	In vitro study	The aim is to create a drug delivery system by coating titanium foils with graphene oxide and titanate, with the goal of boosting the growth and differentiation of rBMSCs towards osteogenesis.	GO sheets, generated using a modified Hummer’s method, were integrated with bioactive titanate onto titanium implants (referred to as GO-Ti) prior to reduction (resulting in rGO-Ti). The growth of rBMSCs on these surfaces was evaluated through mRNA expression and alkaline phosphatase activity.	The findings demonstrated excellent performance of the Dexamethasone-loaded surface (DEX-GO-Ti) in promoting cell proliferation. On DEX-GO-Ti, significant expression of osteogenic differentiation-related proteins, mRNA, and calcium was observed in RMBSCs.
Park et al. [47] 2023	In vitro study	Atmospheric pressure plasma was employed to apply a coating of graphene possessing photothermal characteristics onto a zirconia surface.	Utilizing an atmospheric pressure plasma generator (PGS-300, Expantech, Suwon, Republic of Korea), an Ar/CH4 gas combination was applied to a zirconia sample at a power level of 240 W and a flow rate of 10 L/min.	The category where the zirconia sample, covered with graphene oxide, underwent near-infrared ray exposure and exhibited a noteworthy decrease in the attachment of *S. mutans* and *P. gingivalis* in comparison to the non-irradiated group.
Cheng et al. [48] 2022	In vitro study	The aim of the study was to evaluate the antibacterial properties and cytocompatibility of a novel composite coating containing GO and the antimicrobial peptide (AMP) Nal-P-113 on a smooth titanium surface.	Smooth titanium surface coated with GO and antimicrobial peptide (AMP) Nal-P-113.	The Nal-P-113-loaded GO coating exhibited potent antibacterial activity against both Gram-positive (*S. mutans*) and Gram-negative (*P. gingivalis*) bacteria while maintaining biocompatibility with HGF cells.
Jang et al. [49] 2021	In vitro study	To investigate how the application of GO onto a zirconia surface influences the attachment of bacteria and the activation of osteoblasts.	The atmospheric pressure plasma generator (PGS-300) was used to apply a blend of Ar/CH4 gas onto zirconia samples, dividing them into two groups: uncoated (Zr group) and graphene oxide-coated (Zr-GO group).	GO-coated zirconia effectively obstructs *S. mutans* bacteria adhesion, promoting osteoblast growth and specialization. This suggests its potential in combating peri-implantitis by deterring bacterial attachment and enhancing bone adhesion, thereby improving implant success rates.
Guo et al. [50] 2021	In vitro study	To test the antimicrobial effects of PEEK-PDA-GO surfaces	Antibacterial and cellular tests	PEEK-PDA-GO effectively inhibits microorganisms such as Streptococcus mutans, Fusobacterium nucleatum, and Porphyromonas gingivalis, promoting strong human gingival fibroblast adherence and proliferation.
Qin et al. [51] 2021	In vitro study	To test the effect of GO-carbon fibers (CF)-PEE coating on Titanium implants	Physiochemical and cellular tests	GO-CF-PEEK: - Antimicrobial effects. - Reducing the coefficient of friction and improving wear resistance. - Cytocompatibility on murine fibroblasts.
Qin et al. [52] 2021	In vitro and in vivo study	To test biological safety and osteointegration of GO-CF-PEEK coatings.	Cellular tests and in vivo analysis of osseointegration.	GO-CF-PEEK:Surface hydrophilicity was increased. Porous nanostructures improved early cell activities and osseointegration.
Qin et al. [53] 2020	In vitro study	To determine whether polymicrobial biofilms can be removed using GO.	The study examined in vitro biofilm formation on titanium surfaces using brushing alone, varying GO concentrations, combined treatments, and no therapy.	GO at high concentrations removed bacteria and prevented biofilm reformation in combination with brushing (Group GB). The BMSCs’ osteogenic capacity was increased on the GO Ti surfaces.
Patil et al. [54] 2020	In vitro study	To determine the effects of titanium alloy, graphene, and reduced graphene oxide (rGO) on tension and distortion at the implant.	Finite element analysis (FEA).	Titanium implants had better mechanical behavior than graphene when coated with rGO.
Jeong-Woo Kim et al. (2017) [55]	In vivo	To evaluate the effect of biphasic calcium phosphate (BCP) coated with reduced graphene oxide (rGO) as bone graft materials on bone regeneration.	- BCP coated with rGO fabricated at various concentrations. - Cell viability tests conducted at different rGO concentrations.	- New bone formation evaluated using micro-CT and histological analysis. - Effectiveness of rGO-coated BCP on osteogenesis. - Importance of composite concentration.
Erika Nishida et al. (2016) [56]	In vitro	To ascertain whether the graphene oxide scaffold promoted bone induction in the extractive alveoli of dog teeth.	- Fabrication of GO-applied scaffold - and dispersion on collagen sponge scaffold - Characterization using SEM, physical testing, cell seeding, and rat subcutaneous implant testing - Implantation of GO scaffold into dog tooth extraction socket - Histological observations at 2 weeks post-surgery.	- Improved physical strength, enzyme resistance, calcium, and protein adsorption due to GO application. - Increased osteoblastic cell proliferation with GO application. - Good biocompatibility observed through rat subcutaneous tissue response. - Enhanced bone formation in dogs.
Jong Ho Lee et al. (2015) [57]	In vitro	To examine whether reduced graphene oxide (rGO) and hydroxyapatite (HAp) nanocomposites (rGO/HAp NC) could enhance MC3T3-E1 preosteoblast osteogenesis and promote new bone formation.	- Examination of the potential of graphene-based hybrid composites for cellular differentiation and tissue regeneration.	- Synergistic promotion of osteodifferentiation without hindering proliferation observed with rGO/HAp combination - Graphene-based composites found to have osteogenesis stimulation potential.
Izumi Kanayama et al. (2014) [58]	In vitro	To examine the bioactivity of graphene oxide (GO) and Reduced graphene oxide (RGO) films and collagen scaffolds coated with GO and RGO.	- Evaluation of GO and RGO films’ bioactivity and collagen scaffolds coated with GO - Biological properties assessed using SEM, atomic force microscopy, calcium adsorption tests, and MC3T3-E1 cell seeding. - Implantation of scaffolds into rat subcutaneous tissue, followed by DNA content and cell ingrowth measurements 10 days post-surgery.	- GO and RGO films possess distinct biological properties: enhanced calcium adsorption and alkaline phosphatase activity, promoting osteogenic differentiation; - GO- and RGO-coated scaffolds exhibit higher compressive strengths compared to non-coated scaffolds. - RGO-coated scaffolds are more bioactive than GO-coated scaffolds.
Chingis Daulbayev et al. (2022) [59]	In vitro	The GO/HAp composite prepared was dispersed in biodegradable polymer-polycaprolactone (PCL) in order to design a composite scaffold with the aim of enhancing osteogenic differentiation of osteoblasts for potential medical application	- Utilization of biodegradable polycaprolactone (PCL), graphene oxide (GO), and calcium hydroxyapatite (HAp). - Dispersal of GO/HAp composite in PCL for composite scaffold creation aimed at enhancing osteogenic differentiation of osteoblasts.	- Obtained composite scaffold suitable for bone tissue regeneration with antimicrobial properties.
Milena Radunovic et al. (2017) [60]	In vitro	To investigate the biocompatibility of GO-coated collagen membranes on human dental pulp stem cells (DPSCs), focusing on the cytotoxicity of biomaterials and the ability to promote the differentiation process of DPSCs and to control the induction of the inflammatory event.	- Investigation of biocompatibility of GO-coated collagen membranes on human dental pulp stem cells (DPSCs),	- Faster DPSCs differentiation into odontoblasts/osteoblasts induced by GO-coated membranes - Potential as an alternative to conventional membranes for efficient bone formation and improved clinical performance.
Letizia Ferroni et al. (2022) [61]	In vitro	The amount of Rgo filler was defined to achieve a biocompatible and antibacterial PCL-based surface that supports human mesenchymal stem cell (MSC) adhesion and differentiation. Compounds containing three different percentages of Rgo were tested.	- Evaluation of PCL-based surfaces with reduced graphene oxide (Rgo) nanofillers for bone regeneration in dentistry.	- Different percentages of rGO filler in PCL tested for biocompatibility and antibacterial properties - All scaffolds exhibit biocompatibility, antibacterial properties, adhesion, and differentiation of human mesenchymal stem cells (MSCs).
Elham-Sadat Motiee et al. (2023) [62]	In vitro	Poly-3 hydroxybutyrate-chitosan (PC) scaffolds reinforced with graphene oxide (GO) were fabricated by the electrospinning method to evaluate the possible increase in the biomechanical properties of the scaffolds.	- Development of Poly-3 hydroxybutyrate-chitosan (PC) scaffolds reinforced with graphene oxide (GO) via electrospinning method - Investigation of how GO reinforcement affects fibers diameter, thermal capacity, surface hydrophilicity, mechanical properties, and degradation of the nanocomposite scaffolds.	Improved physicochemical, mechanical, and biological properties demonstrate the potential of PCG nanocomposite scaffolds for bone tissue engineering.
Alana P C Souza et al. (2022) [63]	In vitro	Develop a chitosan-xanthan (CX) membrane associated with hydroxyapatite (HA) and different concentrations of graphene oxide (GO).	The study developed a chitosan-xanthan membrane with HA and GO concentrations, characterized using various techniques, including X-ray diffraction, FTIR, Raman spectroscopy, SEM, contact angle, tensile strength, bioactivity, and cell viability.	- Membranes with non-porous, homogeneous surfaces, hydrophilic nature, higher tensile strength, and reliability for guided bone regeneration therapies are essential for various applications.

## Data Availability

Not applicable.

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
