# Peer review of "Application of Graphene Oxide in Oral Surgery: A Systematic Review"

_materials, 2023, doi:10.3390/ma16186293_

Round 1

Reviewer 1 Report

The article summarizes in a very appropriate way the use of graphene in oral surgery.

However, the article presents some improvements that may be interesting:

First, the inclusion and exclusion criteria must be indicated in the abstract

In the aspects related to the search by keywords, we speak of dental Surgery and Dentistry but in no case of Oral Surgery. Therefore, an exclusion criterion should be added in which articles that do not focus on oral surgery should be eliminated since there will be some articles that focus on other branches of dentistry

The conclusions need to be summarized further. It's a very long paragraph.

Is figure two owned by the authors or has it been collected from another article?

Author Response

Dear Reviewer,

Thank you for carefully reading our manuscript. Thanks for your suggestion. Changes in manuscript are in yellow. Below our responses to your answers.

  1. First, the inclusion and exclusion criteria must be indicated in the abstract.

Thank you for your suggestion. We added the inclusion and exclusion criteria in the abstract: “Reviewers worked in pairs to select studies based on specific inclusion and exclusion criteria. They included animal studies, clinical studies or case reports, and in vitro and in vivo studies. However, they excluded systematic reviews, narrative reviews, and meta-analyses.”

  1. In the aspects related to the search by keywords, we speak of dental Surgery and Dentistry but in no case of Oral Surgery. Therefore, an exclusion criterion should be added in which articles that do not focus on oral surgery should be eliminated since there will be some articles that focus on other branches of dentistry.

Thanks for the suggestion but oral surgery is often referred to as dental surgery which includes procedures such as implant placement, bone regeneration and other maxillomandibular surgeries, all topics addressed in the discussion of that article.

  1. The conclusions need to be summarized further. It's a very long paragraph.

Thank you for your suggestion. We have summarized the conclusions: “The field of graphene-based engineered nanomaterials in dentistry is expanding due to their superior mechanical properties, electrical conductivity, and thermal stability. Graphene and its derivatives can be functionalized with bioactive compounds and added to dental materials, enhancing their properties. These materials stimulate tissue regeneration, cell differentiation, and proliferation while being biocompatible and low in toxicity. Dental implants and abutments with graphene coatings exhibit improved cytocompatibility, antibacterial properties, and osteoblast growth.

Graphene-enhanced scaffolds and membranes for guided bone regeneration (GBR) al-so have improved physical properties, leading to enhanced bone formation. Laboratory tests indicate increased secretion of osteogenic markers and reduced inflammatory markers on graphene-coated materials. However, there are concerns about the safety of graphene and its derivatives, necessitating further research to understand their long-term effects. Overall, graphene materials hold great potential for improving oral surgery procedures in the future.”

  1. Is figure two owned by the authors or has it been collected from another article?

Figure 2 was created by authors.

Bari, 14/09/23

Author Response

Title Manuscript: Application of Graphene Oxide in Oral Surgery: A Systematic Review

Manuscript ID: materials-2580315

Dear Reviewer,

Thank you for carefully reading our manuscript: Application of Graphene Oxide in Oral Surgery: A Systematic Review

Thanks for your suggestion. Changes in manuscript are in yellow.

The paper presents the state-of-the-art graphene-based dental surgery applications including implant surfaces and prosthetic abutment surfaces, membranes and scaffolds used in Guided Bone Regeneration procedures.

Comments:

  1. Table 1 is unnecessary, as the details of search/ indicators are already explained in the main text. Thank you for your suggestion, It has been delated
  2. The information presented in Table 3 should be presented more briefly, as the current form is difficult to follow. Thank you for your suggestion, It has been done.

  1. Although only 19 papers were selected for reviewing the main outcomes of GO application in oral surgery, there are many other papers included into the discussions section and reference list, most of them without any significance related to the title. How can the authors explain this? Thanks for your observation. The articles included, in addition to those that passed the eligibility stage, all have some useful references for our review on the use of ghaphene, as it is such a current and broad topic with multiple and innumerable fields of application.

Bari, 14/09/23                                          

Reviewer 3 Report

Dear Authors,

I have had the opportunity to review your systematic review manuscript titled "Application of Graphene Oxide in Oral Surgery: A Systematic Review," submitted to [Journal's Name] under the Special Issue on Advanced Graphene and Graphene Oxide Materials. First, let me commend you for your comprehensive and rigorous approach to this important subject matter.

Strengths:

Scope and Focus: The manuscript is well-conceived, covering the current applications of graphene oxide in dental surgery in a comprehensive way. Your exploration spans from its potential for better osseointegration to its potential role in guided bone regeneration (GBR), making valuable contributions to the field of regenerative dentistry.

Methodological Rigor: The robustness of your search strategy and the transparency in defining inclusion and exclusion criteria enhance the credibility of your findings.

Clinical Implications: Your discussion on the potential clinical applications of graphene oxide in the areas of antibacterial properties and cytocompatibility is particularly noteworthy.

Acknowledgments of Limitations: Your manuscript benefits from a candid discussion of limitations and the need for further research, which speaks to its academic integrity.

Suggestions for improvement:

Text Clarity and Conciseness: Although the manuscript is largely well written, a thorough review is recommended to eliminate unnecessary repetitions and improve text clarity. The manuscript would benefit from being concise to facilitate a smoother reading experience for the audience.

In conclusion, your manuscript constitutes an important addition to the existing literature on the application of graphene-based materials in oral surgery.

Although the manuscript is largely well written, a thorough review is recommended to eliminate unnecessary repetitions and improve text clarity. The manuscript would benefit from being concise to facilitate a smoother reading experience for the audience.

Author Response

Title Manuscript: Application of Graphene Oxide in Oral Surgery: A Systematic Review

Manuscript ID: materials-2580315

Dear Reviewer,

Thank you for carefully reading our manuscript. Thanks for your suggestion. Changes in manuscript are in yellow.

Although the manuscript is largely well written, a thorough review is recommended to eliminate unnecessary repetitions and improve text clarity. The manuscript would benefit from being concise to facilitate a smoother reading experience for the audience.

Thank you for your suggestion. Some repetitions have been eliminated and some sentences in the text have been modified.

Bari, 14/09/23                                          

Round 2

Reviewer 2 Report

The revised version of the manuscript is suitable for publication.